# Study on the Evaluation Method of Sound Phase Cloud Maps Based on an Improved YOLOv4 Algorithm

**DOI:** 10.3390/s20154314

**Published:** 2020-08-02

**Authors:** Qinfeng Zhu, Huifeng Zheng, Yuebing Wang, Yonggang Cao, Shixu Guo

**Affiliations:** Key Laboratory of Acoustics Research, China Jiliang University, Hangzhou 310018, China; zhu632193140@gmail.com (Q.Z.); wyb_1963@163.com (Y.W.); 15A0202111@cjlu.edu.cn (Y.C.); guoshixu@126.com (S.G.)

**Keywords:** YOLOv4 algorithm, sound source localization, sound imaging instrument, positioning error, DenseNet, k-medians++

## Abstract

Most sound imaging instruments are currently used as measurement tools which can provide quantitative data, however, a uniform method to directly and comprehensively evaluate the results of combining acoustic and optical images is not available. Therefore, in this study, we define a localization error index for sound imaging instruments, and propose an acoustic phase cloud map evaluation method based on an improved YOLOv4 algorithm to directly and objectively evaluate the sound source localization results of a sound imaging instrument. The evaluation method begins with the image augmentation of acoustic phase cloud maps obtained from the different tests of a sound imaging instrument to produce the dataset required for training the convolutional network. Subsequently, we combine DenseNet with existing clustering algorithms to improve the YOLOv4 algorithm to train the neural network for easier feature extraction. The trained neural network is then used to localize the target sound source and its pseudo-color map in the acoustic phase cloud map to obtain a pixel-level localization error. Finally, a standard chessboard grid is used to obtain the proportional relationship between the size of the acoustic phase cloud map and the actual physical space distance; then, the true lateral and longitudinal positioning error of sound imaging instrument can be obtained. Experimental results show that the mean average precision of the improved YOLOv4 algorithm in acoustic phase cloud map detection is 96.3%, the F1-score is 95.2%, and detection speed is up to 34.6 fps. The improved algorithm can rapidly and accurately determine the positioning error of sound imaging instrument, which can be used to analyze and evaluate the positioning performance of sound imaging instrument.

## 1. Introduction

Sound imaging instruments, also known as acoustic cameras, are a special kind of acoustic analysis equipment that use a microphone array to measure the distribution of the sound field in a certain spatial range. A sound imaging instrument can be used to measure the position of the sound source and the state of sound radiation as well as to display visual images with the aid of a cloud diagram, which is widely used in the research field of sound source positioning.

Research on sound source localization began in 1996, when Silverman and Brandstein [1] first performed sound source localization experiments using microphone arrays. Subsequently, in 2000, Asano et al. [2] used large-scale microphone arrays to locate the glide noise of a Boeing 777, enabling the real-time monitoring and analysis of moving object noise. In 2006 Li and Chen [3] developed an acoustic field visualization system consisting of microphone arrays to perform acoustic signal acquisition and processing as well as sound source localization. However, no criteria exist for evaluating the results of sound source identification and localization systems. Not until 2014 did the Chinese General Administration of Quality Supervision, Inspection and Quarantine issue the “Sound Source Identification and Localization System Calibration Specification,” which provides only preliminary technical standards for the quantitative evaluation of sound source localization systems and is applicable only to sound source localization systems based on the beamforming algorithm. Sound imaging instruments, which are a part of sound source localization systems, are acoustic devices that combine microphone array technology with digital camera technology to achieve sound source localization in the form of cloud mapping while displaying real scene sound sources [4,5]. No uniform measurement technique is available for assessing source localization imaging results in acoustic phase cloud maps. Also, there may be some differences in the array elements on the microphone array during the manufacturing process of the sound imaging instruments, and such errors can have a serious impact on the positioning of the sound source. Traditional acoustic calibration methods [6] can only calibrate the measurement characteristics of a single array element to eliminate the differences between microphone array elements. However, the errors caused by inaccurate element positions cannot be calibrated [7]. Therefore, not only is a sound imaging instrument’s calibration scheme needed, but also a suitable method should be developed to assess source localization results in cloud maps.

The technical parameters and calibration methods of sound imaging instruments are provided by manufacturers and are not the same. The calibration of acoustic phasers by manufacturers is generally divided into two categories. The first type corrects the system errors by setting parameters through manual measurements. In the second category, the image processing method extracts the detection object on the acoustic phase cloud map and compares it with the actual target position to obtain its localization error. To some extent the image processing method saves the time required for manual measurements, but when the detection background is complex, extracting the edges of the inspected object is difficult.

On the basis of the aforementioned issues, this study investigates the calibration of sound imaging instruments in combination with deep learning. The generalizability of features extracted using target detection methods involving deep learning is considerably higher than that of traditional artificial features. Target detection methods involving deep learning have wide applications such as in parts inspection, facial recognition, and automated driving [8]. The main methods can be divided into two types. The first type is a target detection algorithm based on area recommendations, an example of which is the Faster-RCNN algorithm [9]. Another type of detection algorithm is based on regression methods, which treat the detection problem as a regression problem, directly predicting the target location and category, and its speed of operation is more in line with real-time detection requirements compared with the first type of algorithm. An example of the second type is the YOLOv4 [10] algorithm.

In this study, we propose an acoustic phase cloud map evaluation method based on an improved YOLOv4 algorithm, which combines the advantages of the high precision of deep learning and meeting real-time applications and divides sound source localization results into lateral and longitudinal localization errors. This method trains the neural network by using acoustic phase cloud maps generated by sound imaging instruments to locate sound sources at different positions as a dataset. The sound phase cloud map is fed into the improved YOLOv4 network to obtain the image region of the target sound source and a pseudo-color map, and the pixel coordinates of the region’s center are calculated. After converting them into physical space coordinates through calibration, lateral and longitudinal positioning errors are calculated.

## 2. Evaluation of Sound Imaging Instrument Positioning Errors

### 2.1. Definition of Positioning Errors

Sound imaging instruments combine electronics and information processing technologies to visualize measurement data. It superimposes the acoustic phase map generated using the microphone array with video images taken by the camera installed on the array in a transparent way to form a so-called acoustic phase cloud map, from which the noise state of the measured object can be visually analyzed.

Because of the asymmetry of microphone arrays, the acoustic resolution of sound imaging instruments varies in both lateral and longitudinal directions during the detection of a sound source.

As shown in Figure 1, the microphone array plane is parallel to the sound source trigger plane, and a coordinate system is established with the center of the sound phase image as the origin. $\left(x_{0},y_{0}\right)$ and $\left(x_{1},y_{1}\right)$, respectively, correspond to the two-dimensional coordinate points of the target sound source center and the pseudo-color map center in the sound phase image obtained using the sound imaging instrument. The difference between lateral coordinates is the lateral positioning error, and the difference between longitudinal coordinates is the longitudinal positioning error.

### 2.2. Acquisition of Positioning Errors

The YOLOv3 algorithm is used to locate the target sound source and the pseudo-color map in the acoustic phase cloud map, and the predicted bounding box of both the target sound source and pseudo-color map can be obtained. After averaging the coordinates of the four vertices of the predicted bounding box, the pixel coordinates of the center are obtained. Once obtained, these coordinates are converted into actual position coordinates based on the ratio of the unit pixel to the actual size. According to Zhang’s calibration principle [11], a standard chessboard grid with a known actual size of $L_{0}\times H_{0}$ is placed at the position of the sound source, and the lateral and longitudinal proportions of the sound phase cloud map can be calibrated. It can calculate the lateral scale factor $k_{L}=L_{0}/L_{1}$ and the longitudinal scale factor $k_{H}=H_{0}/H_{1}$ of the image, where $L_{1}$ and $H_{1}$ respectively are the lateral and longitudinal dimensions of the standard chessboard grid in the sound phase cloud map (expressed as the number of pixels). After determining the lateral scale coefficient $k_{L}$ and the longitudinal scale coefficient $k_{H}$ of the acoustic phase cloud map, we can acquire the actual position coordinates of the target sound source and the center of the pseudo-color map as well as the positioning error of the sound imaging instrument. The lateral positioning error is $\alpha_{\varepsilon }=k_{L}|x_{0}-x_{1}|$ and the longitudinal positioning error is $\beta_{\varepsilon }=k_{H}|y_{0}-y_{1}|$.

## 3. Study of the YOLO Algorithm

### 3.1. Principle of the YOLO Algorithm

The YOLOv3 [12] algorithm adds applied improvements such as multiscale detection and multi-label classification to the YOLOv2 [13] algorithm, and uses the improved Darknet53 [12] based on residual neural networks as a feature extractor. This improvement addresses the shortcomings of the YOLO [14] series of algorithms that are not efficient in detecting small objects.YOLOv3 has thus become one of the best target detection algorithms to date. The YOLOv3 algorithm first scales the original image to a size of 416 × 416 and divides the scaled image into 1 × 1 uniformly equal grids based on the size of the original image in relation to the detected target. If the grid contains detection targets, the grid generates B predictive bounding boxes and the corresponding confidence score. Simultaneously, each predicted bounding box is given a probability value for each classification, and the classification in which the probability maximum is located is the category of the bounding box. The confidence level is defined as follows:(1)$$Confidence=p_{r}(\mathrm{Object})\times IoU_{pred}^{truth},p_{r}(\mathrm{Object})\in \{0,1\}$$
where $p_{r}(\mathrm{Object})$ is the probability that the predicted bounding box contains a detection target, and $IoU_{pred}^{truth}$ is the intersection ratio of the target reference bounding box to the predicted bounding box. Finally, bounding boxes predicting the same object are filtered through non-maximal suppression (NMS) [9] to obtain the best bounding box. The prediction process is shown in Figure 2.

The network prediction process generates the sizes 13 × 13, 26 × 26, and 52 × 52 of the three scales of feature maps to detect the target object and uses 2× upsampling to enable the detection process to combine the features of different sizes.

The YOLOv3 algorithm performs convolutional prediction for the generation of three scale feature maps through (5+c)∗B convolutional kernels of size 1 × 1, where B is the number of prediction bounding boxes (taken by default as 3). c is the number of categories of the predicted target, and 5 contains four offset parameters responsible for predicting the target bounding box with the confidence level of the probability of containing the target within the target bounding box. Figure 3 shows the prediction process for the target bounding box.

The dashed region in the figure is the initial bounding box and the solid region is the predicted bounding box after the network iteration parameter, where $\left(c_{x},c_{y}\right)$ and $\left(p_{w},p_{h}\right)$ are the center coordinates and width and height dimensions of the bounding box on the feature map, respectively, whereas $\left(t_{x},t_{y}\right)$ and $\left(t_{w},t_{h}\right)$ represent the network-predicted center offset of the bounding box and the scaling of width and height. The conversion process from the initial bounding box to the predicted bounding box is shown in the formula on the right side of Figure 3, where the conversion function is the sigmoid function.

### 3.2. Parameter Optimization of the Anchor

An anchor is a set of a priori boxes with fixed width and height. In the target detection process, the size of the a priori frame directly affects the accuracy of the detection. Therefore, in neural network training, it is particularly crucial to set anchor parameters according to the inherent characteristics of the detection target. The traditional YOLOv3 algorithm uses a K-means clustering algorithm for target objects, using the Euclidean distance as an indicator of similarity under the condition of randomly initializing clustering centers.

When the dataset X is divided with n samples into k categories and the clustering center is iteratively updated to minimize the squared sum of clusters containing data points in each category, it is termed as minimizing, which is determined as follows:(2)$$J=\sum_{i=1}^{n}\sum_{k=1}^{k}∥x_{i}-u_{k}{∥}^{2}$$

Such methods are prone to put the network into local optimization due to the randomness of the initial clustering center, and the final clustering results will tend to generate large-size prior frames. Randomness will hinder the regression of small-size predictive bounding box parameters in actual training and reduce the network’s accuracy in detecting tiny target objects. To solve the aforementioned problem, this study proposes a K-medians++ clustering algorithm combining the K-means++ [15] and K-medians algorithms.

On the one hand, according to the density of data distribution, the initial clustering center is rational, and points containing more data are selected as the new clustering center. On the other hand, the distance metric is used instead of the Euclidean distance as a similarity metric to update the clustering center based on the median of the distance metric. The distance metric is calculated as follows:(3)$$d_{cen}^{box}=1-IoU_{cen}^{box}$$

In this formula, box is the labeled sample frame, cen is the clustering center, and IoU is the cross-totality ratio.

To adapt to the elemental characteristics of the sound phase cloud map and achieve the optimal training effect, the dimensional clustering analysis of the target sound source and pseudo-color map is performed using K-means, K-medians, K-means++, and the improved K-medians++ clustering algorithm. The average cross-merge ratio of the labeled sample frame to the prior frame is finally used as a criterion to evaluate the clustering effect of the algorithm. The average cross-merge ratio is calculated as follows:(4)$$\mathrm{avg}\text{\_}\mathrm{IoU}=\frac{1}{n}\sum_{box=0}^{n}\sum_{cen=0}^{k}\left[1-\max(IoU_{cen}^{box})\right]$$

## 4. Improved Design and Implementation of the YOLOv4 Algorithm

### 4.1. Improvement of the Network Architecture

A network structure combining YOLOv3, YOLOv4, and DenseNet [16] is constructed to meet the requirements of sound imaging instrument meter testing, as shown in Figure 4. Because of the uniformity in the size of the sound imaging instrument’s exported images, the SPP block [17] on YOLOv4 is removed and a dense block is added in its place, allowing a better transfer of feature information and gradients throughout the network, and mitigating overfitting to some extent. The traditional dense block connected by H1 function contains BN-ReLU [18]-Conv (1 × 1)-BN-ReLU-Conv (3 × 3). In DenseNet, all previous layers are connected as inputs:(5)$$x_{l}=H_{l}\left(\left[x_{1},x_{2},\cdots ,x_{l-1}\right]\right)$$
where $\left[x_{1},x_{2},\cdots ,x_{l-1}\right]$ is the stitching of all feature maps before the layer $x_{l}$. The dense block’s feedforward model combines feature information to perform nonlinear transformation processing, which facilitates feature reuse and considerably reduces the number of parameters.

The use of CSPDarknet53 [19] as the backbone of the network, which combines the residual block [20] in Darknet53 with the CSP model. Compared to the Darknet53 used by YOLOv3, there is a significant decrease in overall network computation, and this part of the performance optimization does not affect the predictive accuracy of the network, even if there is a slight improvement. In the neck part of the network, the path aggregation network (PAN) [21] is used. Compared with FPN [22] used by YOLOv3 for multi-scale feature fusion, PAN shortens the path of high and low fusion and has more flexible ROI pooling. A bottom-up path is also added to make it easier to disseminate information at lower levels. Table 1 shows a quantitative analysis of the variation in network performance due to different blocks.

The detection accuracy obtained from the AP_50_ and AP_75_ corresponding to different thresholds filtering the optimal prediction frame in Table 1. AP_S_, AP_M_, and AP_L_ correspond to the detection accuracy of detecting images of different sizes. As can be seen in Table 1, the introduction of the CSP model improves the detection speed of the network and improves the detection accuracy by a small amount. Replacing the PAN block with the FPN block increases the repeatability of the information on the feature map, obtaining objective accuracy at the expense of a small amount of speed. The algorithm used in this paper removes the SPP blocks from YOLOv4 and adds Dense blocks. This change in the network structure also improves the accuracy of the algorithm. The improvement in accuracy is especially noticeable when inspecting large images.

The network uses the mish [23] function as an activation function aspect. The slight allowance of negative values by the mish function yields a better gradient flow than the traditional relu function does with a hard zero boundary for negative values. As shown in Figure 5.

The smoothing nature of mish efficiently allows information to penetrate deeper into the neural network, resulting in better accuracy and generalization. Also, as the number of network layers increases, networks using mish show a higher test accuracy than those using relu and softplus do, which is more effective for optimizing complex networks. The Mish function is defined as follows:(6)$$f\left(x\right)=x\cdot \tanh\left(softplus\left(x\right)\right)=x\cdot \tanh\left(\ln\left(1+e^{x}\right)\right)$$

In the case of category loss, focal loss [24] is used, which adds a factor to the original cross-entropy loss. The function is cross-entropy loss when γ=0, where if γ>0, focal loss makes the loss values generated by well-classified examples negligible. The focus on difficult, misclassified samples prevents the effect of simple samples on the network and enhances network generalization capabilities. As shown in Figure 6.

DIoU loss [25] is introduced for predicting boundary box regression. Traditional IoU loss [26] has the problem of gradient disappearance when there is no intersection between prediction and target bounding boxes. DIoU loss solves this problem by introducing a metric parameter to measure the distance and proximity of the two boxes in the operation. DIoU loss is defined as follows:(7)$$L_{DIoU}=1-IoU+{\left(\frac{d}{c}\right)}^{2}$$
where d and c are defined as shown in Figure 7.

Subsequently, CIoU loss [25] introduces the measure aspect ratio similarity υ and the average scale parameter α on the basis of DIoU loss, which can achieve better convergence speed and accuracy in predicting bounding box regression. CIoU loss is defined as follows:(8)$$L_{CIoU}=1-IoU+{\left(\frac{d}{c}\right)}^{2}+\alpha \upsilon$$
(9)$$\upsilon =\frac{4}{\pi^{2}}{\left(\arctan\frac{w^{gt}}{h^{gt}}-\arctan\frac{w}{h}\right)}^{2}$$
(10)$$\alpha =\frac{\upsilon }{\left(1-IoU\right)+\upsilon^{\prime }}$$

The term υ in Equation (9) gradually tends to 0 as $\frac{w}{h}$ increases. In target detection using CIOU as a loss function, we want the size of the detection object to be within a relatively regular range. In the dataset used in this article, the ratio of the length to width of the pseudo-color map and the sound source is within $\left[\frac{1}{3},3\right]$, which can ensure that υ plays a role in the loss function. Equation (11) is the gradient representation of υ with respect to w and h:(11)$$\frac{\partial \upsilon }{\partial w}=\frac{8}{\pi^{2}}(\arctan\frac{w^{gt}}{h^{gt}}-\arctan\frac{w}{h})\times \frac{h}{w^{2}+h^{2}}\;\frac{\partial \upsilon }{\partial h}=-\frac{8}{\pi^{2}}(\arctan\frac{w^{gt}}{h^{gt}}-\arctan\frac{w}{h})\times \frac{w}{w^{2}+h^{2}}$$

In Equation (11), when the length and width are [0,1] and the value of $w^{2}+h^{2}$ is small, there is a gradient explosion, so in practice, we replace $\frac{1}{w^{2}+h^{2}}$ with 1.

All in all, CIoU loss with more favorable convergence is used in prediction bounding box regression, whereas DIoU NMS [25] with relatively less computation is used in the selection of the best prediction bounding box.

### 4.2. Cluster Analysis of Datasets

To improve the robustness of network identification, the dataset is obtained from acoustic phase cloud maps from sound imaging instrument meter performance tests in different environments. Base on the experimental data of the full anechoic room, the outdoor snapshot experiment and the semi-anechoic room experiment are added, as shown in Figure 8.

A total of 750 images are selected for the total dataset, including 150 for the outdoor experiment, 150 for the semi-anechoic chamber experiment, and 450 for the full-anechoic chamber experiment. Considering the problem of insufficient resolution of the sound imaging instrument in outdoor tests, outdoor experiments use a car as the target sound source, and semi-anechoic and full-anechoic chamber experiments use a loudspeaker as the target sound source. With the use of image augmentation processes, such as mirroring, color averaging, light and dark adjustment, and blurring, a single data sample is expanded 10 times to the original dataset. This is shown in Figure 9.

The LabelImg software is used to label sound sources and pseudo-color maps in experimental images and organize them into a VOC dataset format. Randomly selected 4500, 1500, and 1500 images from the dataset are used as training, validation, and test sets, respectively.

Because VOC datasets do not contain sound sources, pseudo-color maps, and other detection data, training with the original anchor parameters exert some effect on training time and accuracy. Therefore, it is necessary to re-cluster the label sample frames of the target sound source and the pseudo-color map to obtain more representative anchor parameters for sound imaging instrument’s localization. K-means, K-medians, K-means++, and K-medians++ algorithms were used for the dimensional clustering analysis of detection tags, with different values of the number of the anchor box k. The average cross-tabulation curve is shown in Figure 10.

As can be observed in Figure 10, the average cross-merge ratio of the four algorithms gradually increases with an increase in the value of k, and the clustering effect gradually improves. The initialization of K-means++ and K-medians++ algorithms with the clustering center rule is smoother and more stable than that of the other two algorithms, which reduces the clustering bias to some extent. With the number of clustering centers in the interval from 6 to 18, the K-medians++ algorithm is superior to the K-means++ algorithm. Simultaneously, as shown in Table 2, when the value of k exceeds nine, a clustering prediction box of a similar size exists, resulting in redundancy, thus resulting in the selection of the clustering result as improved anchor parameters when k=9.

### 4.3. Model Training and Performance Comparison

Prior to the training phase, Darknet-53 network parameters pre-trained using the PASCAL VOC dataset are partially migrated to and initialized on the YOLOv4 base network, and use label smoothing [27] to optimize the dataset. The training phase is optimized using a small batch random gradient descent, setting the momentum parameter to 0.9, the initial learning rate to 0.001, the decay coefficient to 0.1, the batch size is 16, and the parameter γ of focal loss is 2. To reduce the likelihood of the network appearing over-fit, the first 1000 batches of training were warm up [18]. The following learning rate was used: $\eta =\eta_{lr}\times {\left(N_{batch}/1000\right)}^{2}$, where $N_{batch}$ is the current number of batches, and the learning rate reaches 0.001 after 1000 batches. The validation set error is then monitored, and the learning rate multiplies the decay coefficient if the current error is not reduced after 100 epochs. The definitions of terms associated with network training are shown in Table 3.

This experiment uses an Intel(R) Core(TM) i7-8700 processor and NVIDIA GeForce RTX 2080 graphics card. CUDA10.0 and cuDNN7.5 are used to accelerate YOLOv4 network training under Ubuntu 16.04 LTS operating system. The trend of loss during practice is shown in Figure 11.

As can be seen from the Figure 11, the validation set loss at the beginning of training decreases gradually with an increase in the number of training iterations, and the error fluctuation range gradually closes in. After 8000 iterations, the trend gradually stabilizes. The loss value of 9000 iterations fluctuates around 0.3 and no longer tends to get smaller, indicating that the ideal training effect is achieved. Compared with the loss on the validation set, the early loss on the training set drops faster and the final stability loss is also smaller, with a fluctuation of 2.8. The training-generated weight file is used to examine the test set images, using different colors to mark the positions of target objects in the test images and noting the corresponding labels and confidence scores. Figure 12 shows actual test results for different sites.

The improved YOLOv4 algorithm can accurately locate target sound sources and pseudo-colors maps in acoustic phase cloud maps and effectively identify even multiple test targets in an image. Table 4 compares the improved YOLOv4 algorithm with other mainstream algorithms based on the four indexes of class recognition accuracy (AP), average precision average value (mAP), F1-score, and FPS.

In terms of average precision, algorithms of the YOLOv4 family and the Faster-RCNN algorithm are more effective than other algorithms. In terms of recognition speed, one-stage target detection algorithms other than the Faster-RCNN algorithm have better scores. YOLOv4 is slightly slower than YOLOv3 but has improved recognition accuracy. The improved YOLO v4 also sacrifices slightly in speed for greater accuracy compared with the unimproved algorithm. Overall, this method achieves satisfactory results in both recognition accuracy and recognition speed, with mAP of 96.3%, an F1-score was 95.2%, and a detection speed was up to 34.6 fps. Compared with the unoptimized YOLOv4 algorithm, mAP is improved by 1.8% and the F1-score is improved by 1.3%. In terms of the average recognition accuracy of each category, because of the interference of the background environment, the recognition effect of the pseudo color image is the best, and AP can reach 97.4%.

## 5. Experiments and Analysis

The experimental system for a sound imaging instrument to locate sound sources mainly includes sound sources (speakers), a sound imaging instrument, power amplifiers, signal generators, and computers, as shown in Figure 13. The positioning of sound imaging instruments for sound sources at different distances is experimentally measured. Three measurement nodes are set, namely 1 m, 2 m, and 3 m, with a signal generator transmitting 2–5 kHz at each node. Seven groups of single-frequency sinewave continuous signals at an interval of 0.5 kHz are amplified by a power amplifier to drive the sound source. Sound imaging instrumentation locates sound sources excited by different frequency signals in real time and saves results in the form of acoustic phase cloud maps on the computer.

In the plane orthogonal to the sound source at a fixed distance from the sound imaging instrument, the sound imaging instrument is used to image the sound source. During the experiment, the center of the acoustic phase cloud map was used as the coordinate origin, which specifies the x-axis direction of the coordinate system as the lateral direction of the image and the y-axis direction as the longitudinal direction of the image. Acoustic phase cloud maps generated by the sound imaging instrument’s localization imaging of sound sources with different frequencies are input into the YOLOv4 network to locate target sound sources and the pseudo-color map area. Figure 14 shows positioning results for fixed distances (1 m, 2 m, and 3 m), respectively.

The pixel coordinates of the target sound source and the center of the pseudo-color map are obtained from the predicted bounding box. The image is calibrated to obtain the coordinates of their physical positions and the sound imaging instrument positioning error is calculated. The experiment uses the improved YOLOv4 algorithm to locate acoustic phase cloud maps obtained in each case with vocal frequencies ranging from 2–5 kHz at distances of 1 m, 2 m, and 3 m. Five repetitive experiments are conducted at each test point, whereas the localization errors of 105 groups of experimental data are calculated, and the localization errors of the improved YOLOv4 algorithm are shown in Figure 15.

By comparing real coordinates measured using the laser rangefinder with coordinates obtained using the YOLOv4 algorithm, the error of the algorithm in locating the sound source and pseudo-color map, respectively, is obtained. As shown in Figure 15, the maximum positioning error of the YOLOv4 network for the target sound source is 0.3332 cm on the X axis and 0.3132 cm on the Y axis. More than 94.28% of the sound source positioning error is within 0.3 cm. Compared with the error in locating the sound source, due to the variable size of the pseudo-color map, the error produced by the YOLOv4 network in locating the pseudo-color map is on the large side, with the maximum positioning error of 0.5745 cm on the X axis and 0.4878 cm on the Y axis. More than 85.71% of the sound source positioning error is within 0.5 cm. Whether locating the target sound source or the pseudo-color map, the improved YOLOv4 algorithm keeps errors to a minimum.

Pseudo-color maps and sound sources can also be extracted by image processing, and their positioning results are shown in Figure 16. The acoustic phase cloud maps are sequentially processed by difference image method, filtering, binarization, erosion expansion, etc. With the maximum blank area preserved, the acquired images are superimposed and identified by different colors finally.

The excessive variation in light intensity in the anechoic chamber makes it very difficult to extract targets from the acoustic phase cloud map, and the extraction results have more pronounced distortions.

Figure 17 shows the comparison of real error and positioning error which obtained using improved YOLOv4 algorithm and image processing method. Groups 1–7, groups 8–14, groups 15–21 correspond to the situation where the audible frequency is 2–5 kHz when the distance is 1 m, 2 m, 3 m.

As shown in Figure 17, the best localization effect is 1 m away from the target source; when the distance between the two increases, the localization error grows. The localization error of sound imaging instrument tends to decrease with an increase in the frequency of the target sound source, and the longitudinal localization error is more obviously affected by frequency. Lateral and longitudinal errors are affected by the different lateral and longitudinal resolutions of sound imaging instrument, and results vary in individual experiments. In particular, the deviation can be up to 5.0414 cm at the lower frequency of the target sound source. Compared to the positioning errors measured using the image processing method, the errors measured using the improved YOLOv4 are much more obvious that the true value is closer to the true value and the error fluctuates more smoothly in both the lateral and longitudinal directions. Figure 18 shows a comparison curve of the errors introduced by the two measured methods.

The error introduced by using the image processing method is significantly higher than that introduced by using improved YOLOv4, with the maximum lateral method error of 2.5870 cm, with a maximum method error of 3.4159 cm in the longitudinal direction. The positioning error measured using the improved YOLOv4 algorithm is close to its actual error. The maximum deviation in the lateral direction is 0.4526 cm at 1 m, 0.5735 cm at 2 m, and 0.6157 cm at 3 m; the maximum deviation in the longitudinal direction is 0.4772 cm at 1 m, 0.7436 cm at 2 m, and 0.3594 cm at 3 m. The evaluation method used in this study is not affected by the measurement distance and the frequency variation of the target sound source, while the error caused by the evaluation method being within a small range.

## 6. Conclusions

In this study, an improved YOLOv4-based acoustic phase cloud map evaluation method is proposed for the accurate evaluation of sound imaging instrument positioning results. First, YOLOv4 is improved in combination with DenseNet, and anchor parameters are optimized using the K-medians++ clustering algorithm. Second, the trained network is used to identify and localize the target sound source and pseudo-color map in the acoustic phase cloud map. Finally, predicted bounding box parameters are obtained from the algorithm to calculate the acoustic phase localization error and the following conclusions are drawn:(1)The proposed K-medians++ clustering algorithm has better performance than the K-means algorithm used in traditional networks. The clustering effect is more conducive to the optimization of anchor parameters.(2)Compared with DenseNet’s improved YOLOv4 network and the network before optimization, mAP is increased by 1.8%, the F1-score is increased by 1.3%, and the detection speed is up to 34.6fps, which can more effectively identify localized sound sources and false color images.(3)The improved YOLOv4 evaluation method is more accurate than the sound imaging instrument’s positioning error measured using image processing methods. During the experiment, the maximum error introduced by the algorithm in sound source localization is 0.3332 cm, and the maximum error introduced in the pseudo color image localization is 0.5745 cm. In this study, the positioning error of the imaging instrument is similar to its actual error, with the maximum lateral deviation of 0.6157 cm and the maximum longitudinal deviation of 0.7436 cm, which can be controlled within 1 cm of the evaluation accuracy.

Considering that sound imaging instrument is suitable for practical projects, such as car siren capturing and environmental noise detection, we will design experiments for outdoor fields. In addition, we will incorporate a semantic segmentation method with higher recognition and positioning resolution into the study to evaluate the localization error of sound imaging instrument more comprehensively and accurately.

## Figures and Tables

**Figure 1 sensors-20-04314-f001:**
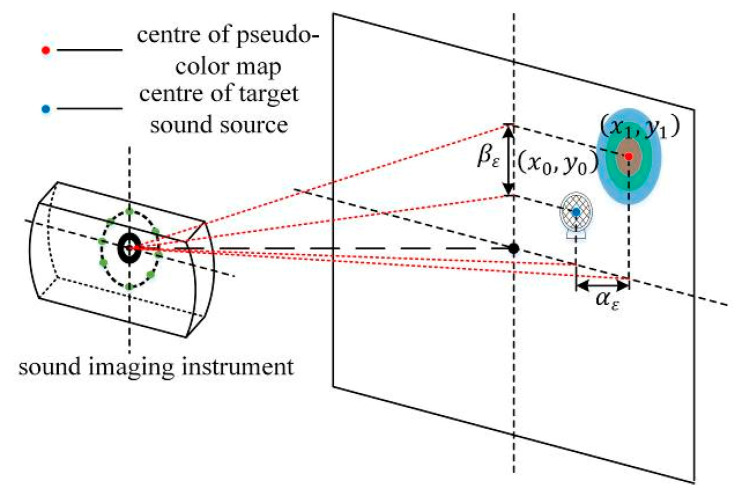
Definition of the positioning error.

**Figure 2 sensors-20-04314-f002:**
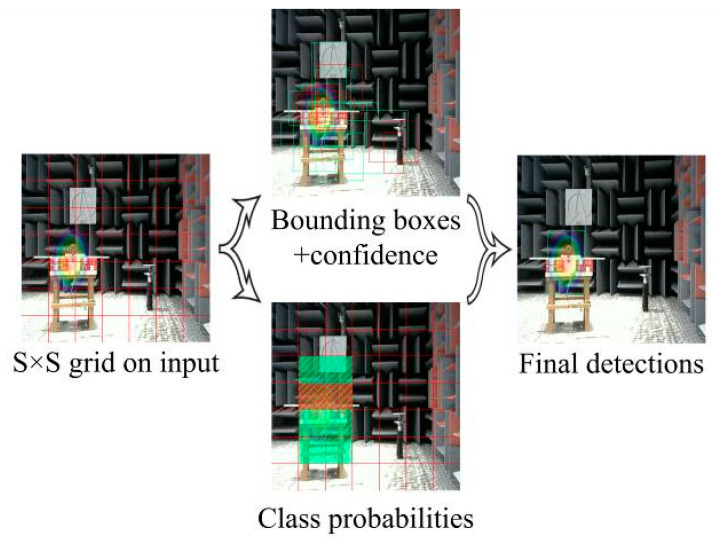
Forecasting process of YOLOv3.

**Figure 3 sensors-20-04314-f003:**
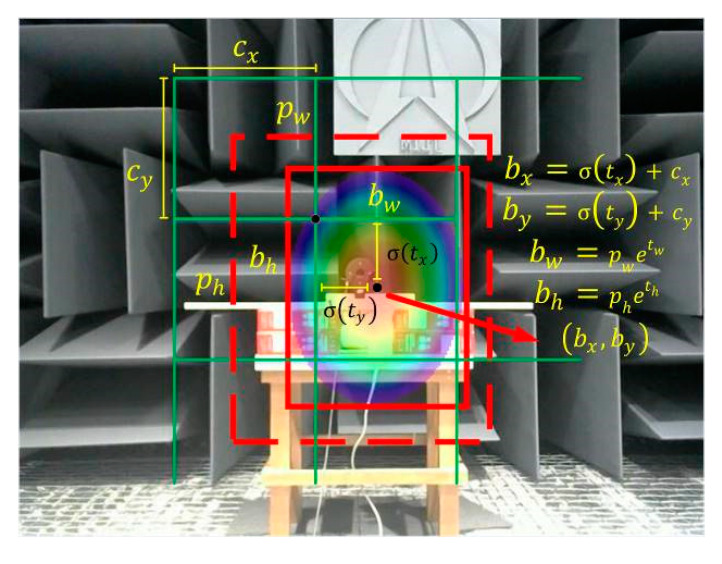
Target bounding box prediction process.

**Figure 4 sensors-20-04314-f004:**
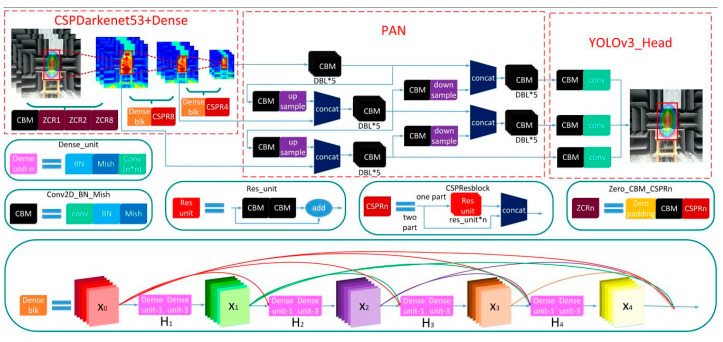
Network structure of improved YOLOv4.

**Figure 5 sensors-20-04314-f005:**
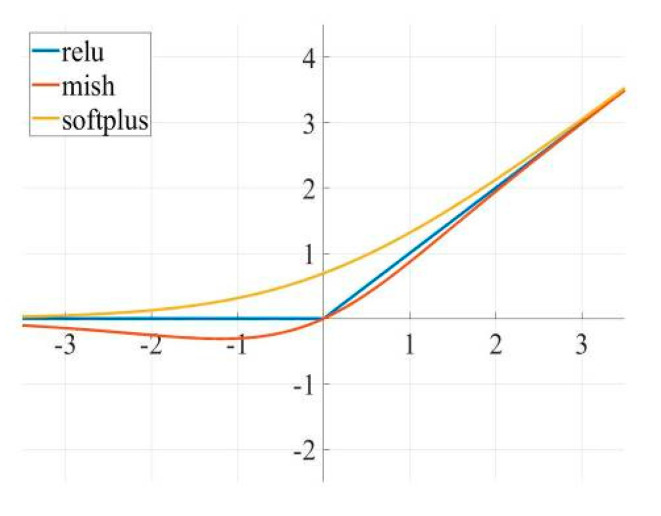
Common Activation Functions.

**Figure 6 sensors-20-04314-f006:**
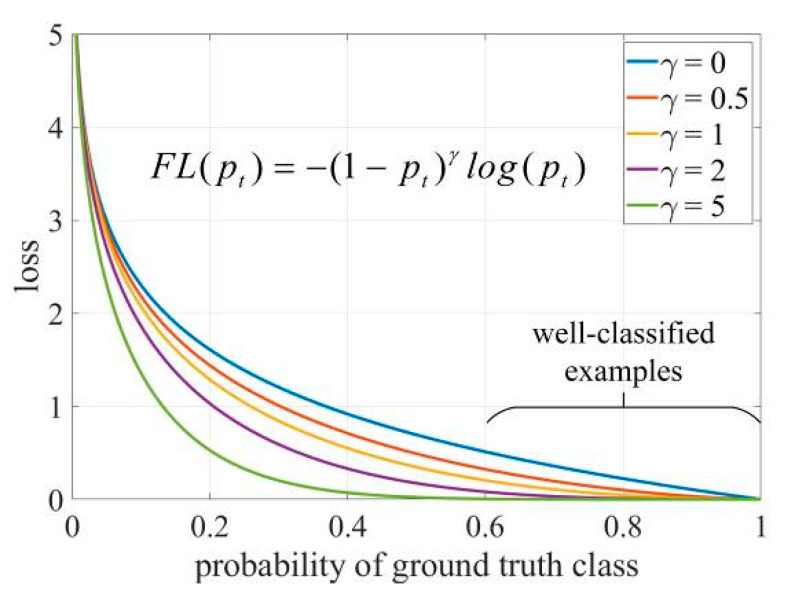
Focal Loss Functions.

**Figure 7 sensors-20-04314-f007:**
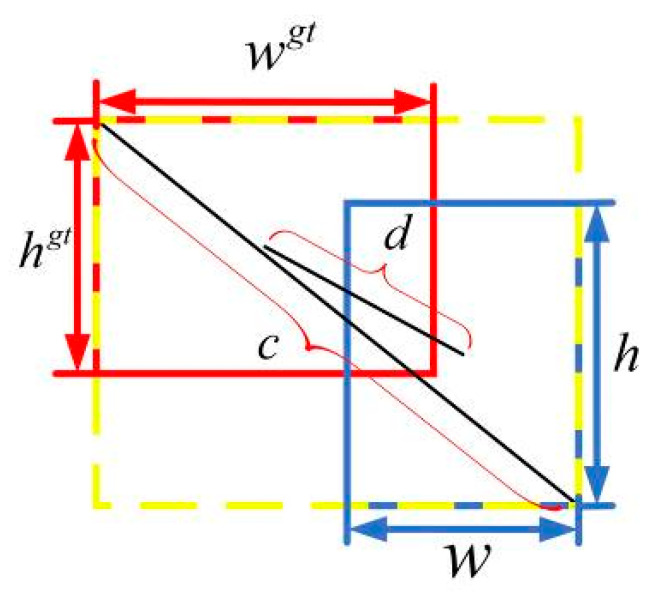
Schematic Diagram of Loss Function.

**Figure 8 sensors-20-04314-f008:**
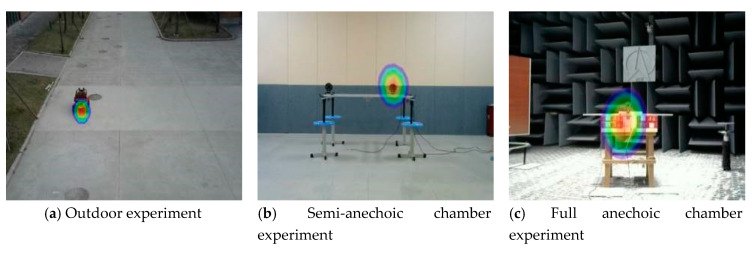
Sound phase image in the data set.

**Figure 9 sensors-20-04314-f009:**
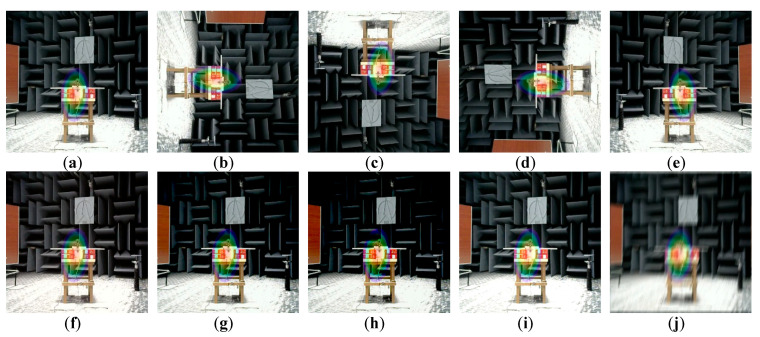
Data set image augmentation: Image augmentation methods: (**a**) original image, (**b**) 90° clockwise rotation, (**c**) 180° clockwise rotation, (**d**) 270° clockwise rotation, (**e**) horizontal mirror, (**f**) color balance processing, (**g**–**i**) brightness transformation, and (**j**) blur processing.

**Figure 10 sensors-20-04314-f010:**
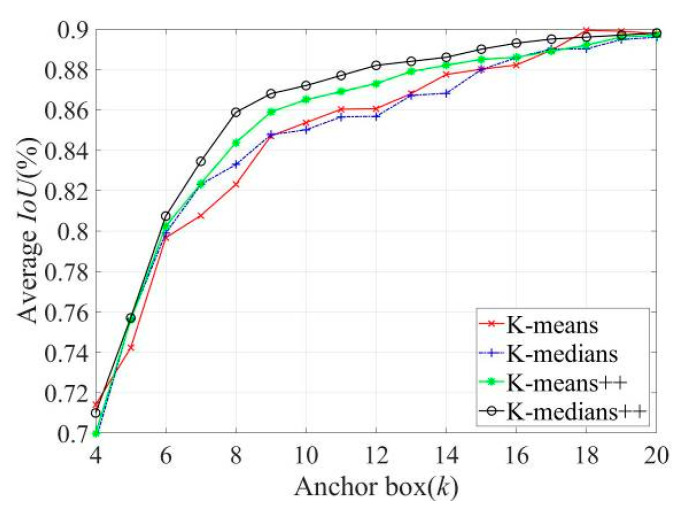
The average curve of the intersection ratio corresponding to different k values.

**Figure 11 sensors-20-04314-f011:**
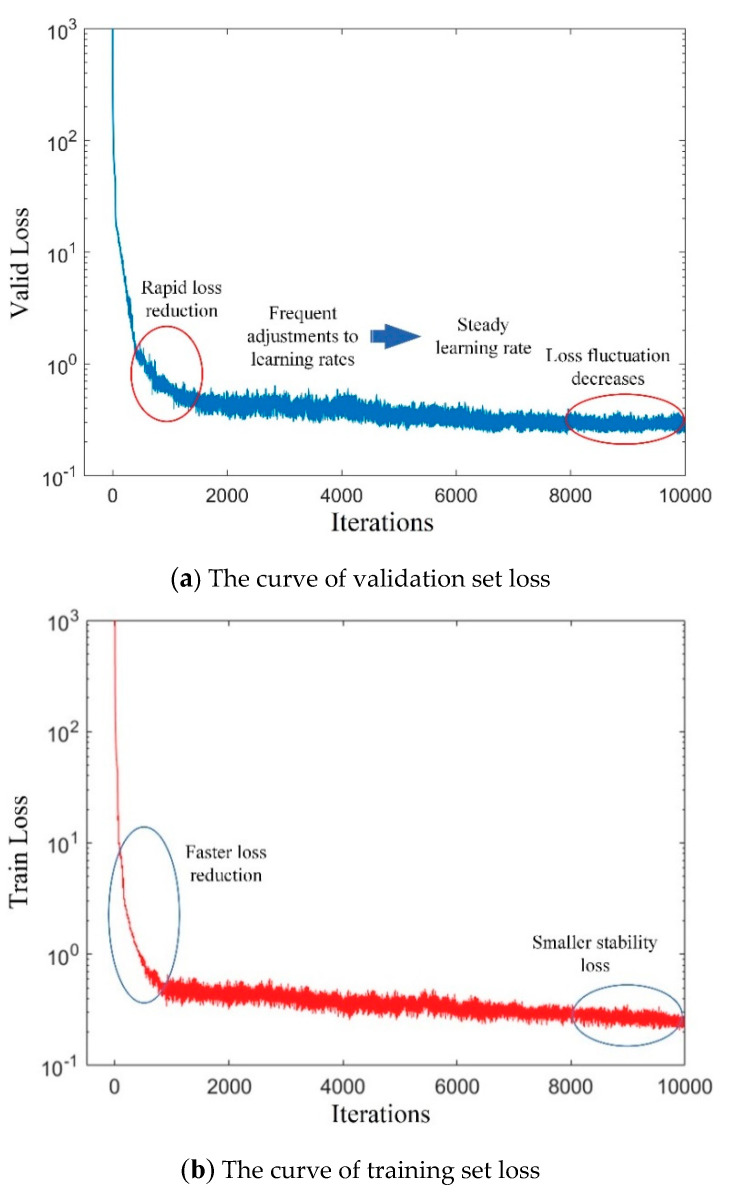
The loss curve.

**Figure 12 sensors-20-04314-f012:**
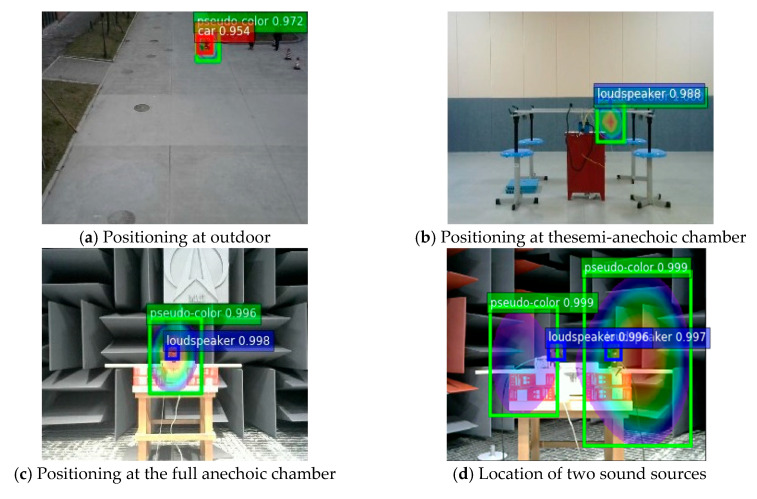
YOLOv4 algorithm positioning results.

**Figure 13 sensors-20-04314-f013:**
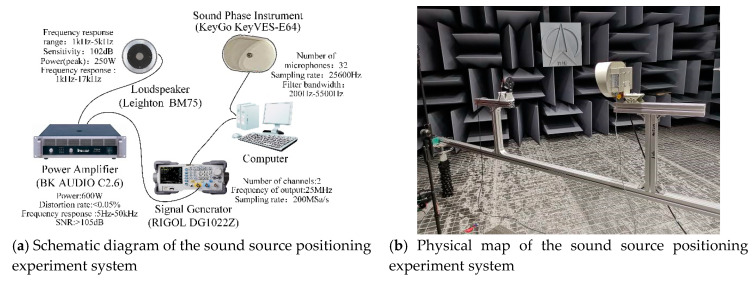
Sound source positioning experiment system.

**Figure 14 sensors-20-04314-f014:**
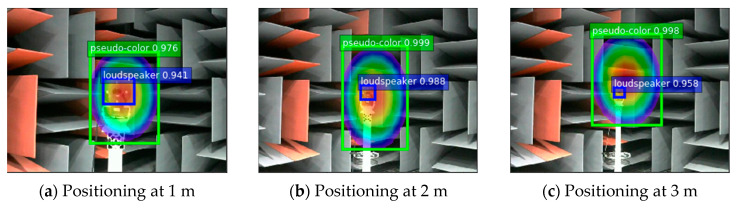
Experimental positioning result.

**Figure 15 sensors-20-04314-f015:**
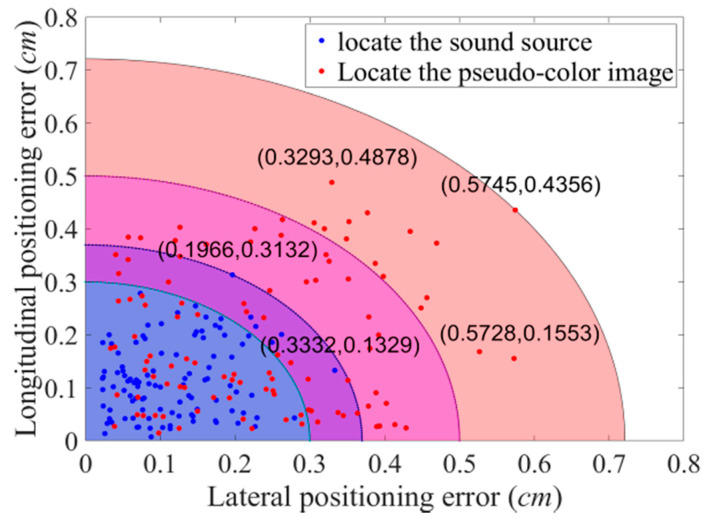
Positioning error of the algorithm in this study.

**Figure 16 sensors-20-04314-f016:**
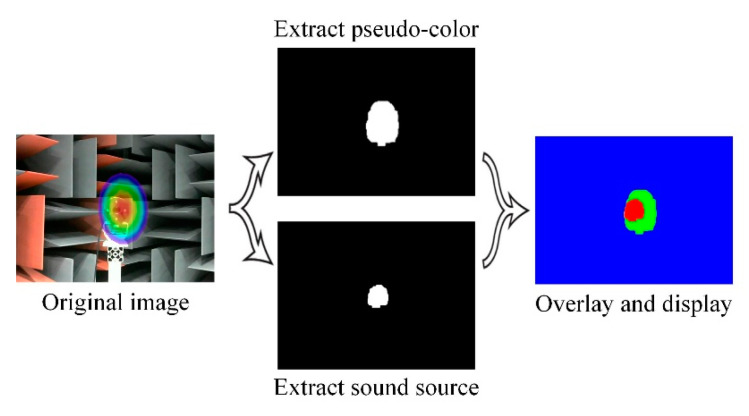
Image processing positioning results.

**Figure 17 sensors-20-04314-f017:**
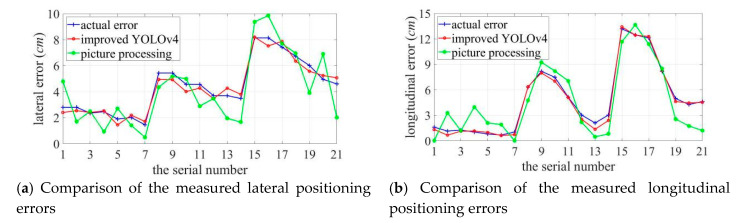
Comparison of the measured positioning errors.

**Figure 18 sensors-20-04314-f018:**
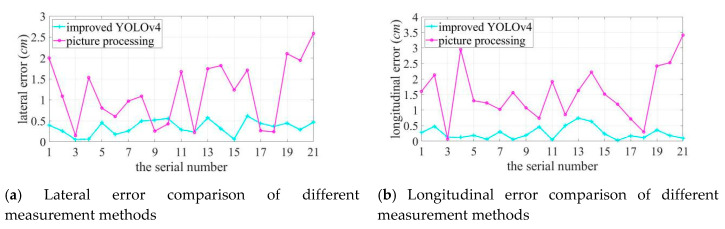
The error comparison of different methods.

**Table 1 sensors-20-04314-t001:** Variation in network performance due to different blocks.

Method	Structure	AP	AP_50_	AP_75_	AP_S_	AP_M_	AP_L_	FPS
YOLOv3	Darkenet53-FPN	30.6%	55.0%	32.1%	15.1%	32.8%	42.6%	35.3
YOLOv3	CSPDarkenet53-FPN	30.8%	55.3%	32.1%	15.0%	32.9%	42.8%	35.6
YOLOv3	Darkenet53-PAN	32.2%	57.6%	34.9%	17.6%	37.2%	49.3%	35.2
YOLOv4	CSPDarkenet53-PAN	40.7%	62.3%	44.0%	20.1%	43.9%	55.7%	34.7
YOLOv4	CSPDarkenet53-Dense-PAN	41.5%	63.7%	45.1%	20.5%	44.8%	57.4%	34.4

**Table 2 sensors-20-04314-t002:** Clustering prediction boxes corresponding to different k values.

k=6	k=7	k=8	k=9	k=10	k=11	k=12
(23,23)	(23,22)	(26,28)	(23,23)	(25,28)	(26,29)	(22,22)
(28,30)	(28,32)	(42,41)	(29,30)	(38,48)	(43,40)	(24,25)
(43,41)	(42,41)	(50,67)	(38,48)	(43,40)	(47,66)	(29,30)
(53,72)	(49,64)	(75,80)	(43,40)	(46,69)	(51,53)	(36,45)
(123,172)	(76,84)	(108,154)	(50,67)	(48,49)	(51,74)	(43,40)
(230,309)	(128,181)	(149,211)	(75,74)	(48,62)	(67,93)	(47,63)
-	(237,319)	(214,281)	(80,92)	(52,54)	(75,74)	(47,71)
-	-	(280,382)	(125,174)	(52,74)	(84,86)	(52,53)
-	-	-	(230,309)	(79,86)	(110,160)	(52, 74)
-	-	-	-	(202,271)	(171,244)	(75,80)
-	-	-	-	-	(247,339)	(125,174)
-	-	-	-	-	-	(230,309)

**Table 3 sensors-20-04314-t003:** Definition of terms related to network training.

Terms Related to Network Training	Definition
Batch	The input pictures for a single network training session. Due to limited computer performance, a single training session of the network does not use all the data in the dataset at once, but rather trains in several batches.
Epoch	When a complete dataset passes through the neural network once and is returned once, this process is called an epoch
Iteration	One iteration is equal to training the network once using a sample of batches.

**Table 4 sensors-20-04314-t004:** Performance comparison of different network structures.

Network Infrastructure	AP-Car	AP-Loudspeaker	AP-Pseudo-Color	mAP	F1-Score	FPS
SSD [28]	89.8%	90.7%	92.7%	91.1%	90.2%	29.4
Faster-RCNN	92.2%	93.4%	95.3%	93.6%	92.4%	10.7
YOLOv3	88.5%	90.8%	91.3%	90.2%	88.9%	35.3
YOLOv4	92.7%	95.1%	95.6%	94.5%	93.9%	34.8
Improved YOLOv4	95.5%	95.9%	97.4%	96.3%	95.2%	34.6

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
