# Peer review of "Study on the Evaluation Method of Sound Phase Cloud Maps Based on an Improved YOLOv4 Algorithm"

_sensors, 2020, doi:10.3390/s20154314_

Round 1

Reviewer 1 Report

The authors present an approach to evaluate the source localization results of the sound imaging instrument by using the widely adopted deep learning algorithm—YOLOv4. Experiments are performed with a loudspeaker to check the algorithm in locating the sound source and pseudo-color map. The positioning error obtained using the improved YOLOv4 algorithm seems to have the same accuracy of the physical evaluation method (using laser). The topic is interesting and the results are well presented. There are several problems that should be addressed in the following issues for the future revision.

1 The English writing should be improved. Long sentences are hard to read. For example, only one long sentence consists of the last paragraph. In addition, the manuscript is too long and can be made more concise.

2 Introduction section, the importance of calibrating the acoustic phasers is missing. The introduction of current technique for calibrating the sound imaging instrument is too simple. No related references are provided.  

3.When introducing the YOLO algorithm, YOLOv3 is referenced. However, the YOLOv4 is used in the study.

4 Details about the training is confusing. For example, batches, epochs, and iterations are used to describe the duration of network training in 4.3. It is better to give out a unified explanation with clear definition. Besides, the samples are used in batches, is it means that the input pictures are fed in batches or the YOLO heads are trained by batched regions sampled from the backbone? what is the batch size? Moreover, the focal loss is used, and what is the γ in the experiments?

5 Figure 12, the information provided is insufficient about the training. There is only a single curve of loss (is it training loss or validating loss?). The training & validating error should be included.

6 According to the description in 4.3, ‘the learning rate multiplies the decay coefficient if the current error is not reduced after 100 epochs’ (which parameter is used here, the training error or validating error?), but in figure 12, it is said that the lr is adjusted more frequently at early stages, where the loss reduces rapidly and the error may also reduce very fast. Is that a mistake?

7 Experimental Procedures section, please provide the detailed information of the experiment setup, especially the model of the employed sound imaging instrument.

8 For a simple sound source, for example, a loudspeaker in Fig.1, the sound source center and the pseudo-color map center should coincide in theory. Positioning error is an actual problem that can be evaluated by the proposed method. However, for a local sound source equipped in a large structure, the engine of a real car, the bounding box of the real target is larger than the bounding box of the pseudo-color map. Is the proposed method still applicable? Or how to confine the bounding box to a part of complex structure?

Reviewer 2 Report

In the present manuscript the authors describe the improvements made on a detection algorithm already in literature for sound imaging source localization. Albeit the introduction is quite short, further details on the background material are provided in the following sections which are effective in conveying the useful information and references to the reader. As a minor side note, the symbols "a" and "b" cited in the text do not appear in formula (1).

The improvement with respect to the state-of-the-art is discussed in the central part of the manuscript: most of the description is concentrated on the activation and loss function. Too few space is dedicated to the improvement of the network structure: this part should be improved by quantifying the effects of the newly introduced blocks, since only a qualitative description is provided. Also, with respect to equation (9), as a reviewer I would like to point out that the right-hand side does not behave well when w is much greater than h: author should comment about eventual ranges and limitations imposed to the arguments of the arctan function. Finally, concerning equation (10), the symbol ν´ (derivative of nu) is not defined; in particular it is not defined what is the derivation variable.

The description of the experiments is well done and the discussion of the results is sound and well-supported. Some language errors are present in the text and should be corrected; a brief list of them follows.

p.1 imaging instrument are > imaging instruments are
p.1 imaging instrument, and > imagin instruments, and
p.1 results of sound > results of a sound
p.1 test of sound > test of a sound
p.1 Sound imaging instrument, also > A sound imagin instruments, also
p.1 is a special > is a piece of special
p.1 range. Sound > range. A sound
p.2 localizations system > localization systems
p.2 mapping, while > mapping while
p.2 Main methods > The main methods
p.2 is Faster-RCNN > is the Faster-RCNN
p.2 region center > region's center
p.2 In addition, no uniform > Also, no uniform
p.2 On the basis of the > Based on the
p.3 instrument varies > instruments varies
p.7 the size of sound > the size of the sound
p.7 allowing better > allowing a better
p.7 feedforward mode > feedforward model
p.8 In addition, as > Also, as
p.8 highter > higher
p.8 become negligible > negligible
p.9 hemi-anhechoic > semi-anhechoic
p.10 hemi-anhechoic > semi-anhechoic
p.10 in order to obtain > to obtain
p.13 system for sound > system for a sound
p.13 (speakers), sound > (speakers), a sound
p.13 maps in the > maps on the
p.13 nodes ate set > nodes are set
p.16 is 1 0.4772 cm > is 0.4722 cm
p.17 error of imaging > error of the imaging

All-in-all the manuscript is of good quality and is worthy to be considered for publications after some minor revision.
